# Using the National Trauma Data Bank (NTDB) and machine learning to predict trauma patient mortality at admission

**Evan J. Tsiklidis[1], Carrie Sims[2], Talid Sinno[1], Scott L. Diamond[1] ***

**1** Department of Chemical and Biomolecular Engineering, Institute for Medicine and Engineering, University of Pennsylvania, Philadelphia, Pennsylvania, United States of America, **2** Department of Trauma Surgery, University of Pennsylvania, Philadelphia, Pennsylvania, United States of America

* sld@seas.upenn.edu

**Data Availability Statement:** Data are available for purchase from the American College of Surgeons (ACS) via https://www.facs.org/quality-programs/trauma/tqp/center-programs/ntdb/datasets. The

## Abstract

A 400-estimator gradient boosting classifier was trained to predict survival probabilities of trauma patients. The National Trauma Data Bank (NTDB) provided 799233 complete patient records (778303 survivors and 20930 deaths) each containing 32 features, a number further reduced to only 8 features via the permutation importance method. Importantly, the 8 features can all be readily determined at admission: systolic blood pressure, heart rate, respiratory rate, temperature, oxygen saturation, gender, age and Glasgow coma score. Since death was rare, a rebalanced training set was used to train the model. The model is able to predict a survival probability for any trauma patient and accurately distinguish between a *deceased* and *survived* patient in 92.4% of all cases. Partial dependence curves ($P_{survival}$ vs. feature value) obtained from the trained model revealed the global importance of Glasgow coma score, age, and systolic blood pressure while pulse rate, respiratory rate, temperature, oxygen saturation, and gender had more subtle single variable influences. Shapley values, which measure the relative contribution of each of the 8 features to individual patient risk, were computed for several patients and were able to quantify patient-specific warning signs. Using the NTDB to sample across numerous patient traumas and hospital protocols, the trained model and Shapley values rapidly provides quantitative insight into which combination of variables in an 8-dimensional space contributed most to each trauma patient's predicted global risk of death upon emergency room admission.

## 1. Background

Trauma is the third leading cause of mortality in the United States and results in approximately 6 million deaths and a cost over 500 billion dollars worldwide each year [1, 2]. A unique characteristic of trauma in relation to other diseased states is not only the large patient-to-patient variability, but non-physiological considerations such as the distance from a trauma center, the resources available for resuscitation, and the number of other casualties. These additional complexities make patient risk analysis difficult, but necessary, to implement in real

authors did not have any special access privileges to this data.

**Funding:** This work was supported by NIH UO1-HL-131053 (S.L.D, T.S.) Funder website: https://report.nih.gov/ Funder Information: Andrei Kindzelski (kindzelskial@nhlbi.nih.gov) The funders had no role in study design, data collection and analysis, decision to publish, or preparation of the manuscript.

**Competing interests:** The authors have declared that no competing interests exist.

time. Given the intricacy of traumatic injury, patient-scale modeling of trauma from first principles is extremely challenging. Consequently, machine learning approaches have been the mainstay of modeling in this arena. In this regard, a large and diverse data set is valuable for the training of an accurate model to efficiently predict patient risk, warning signs, and survival probabilities from easily measurable or estimable quantities.

Trauma centers prioritize patients as they arrive by dividing them into various tiers based upon patient vital signs (respiratory rate, systolic blood pressure, etc.), nature of the injury (i.e., penetrating), and Glasgow Coma score [3]. This prioritization is essential, as it is understood that the sooner a patient receives surgical or medical treatment the greater the likelihood for patient survival [4]. A well-trained model has the potential to help in this patient prioritization by providing a quantitative metric for patient risk. Moreover, the use of SHAP values [5], for example, to explain the prediction of the model may help alert the clinician of difficult-to-discern combinatorial risks in a high dimensional pathophysiological space.

To date, neural networks have been the primary model in the study of trauma. Edwards and Diringer et al. showed that a neural network could accurately classify mortality in 81 intracerebral hemorrhage patients [6]. Marble et al. used a neural network to predict the onset of sepsis in blunt injury trauma patients with 100% sensitivity and 96.5% specificity [7]. Estahbanati and Bouduhi used neural networks to predict mortality in burn patients to a 90% training set accuracy [8]. DiRusso et al. compared the accuracy of logistic regression (linear) and neural networks (non-linear) in predicting outcomes in pediatric trauma patients [9]. Walczak used neural networks to predict the transfusion requirements of trauma patients, an important problem considering potential resource limitations and adverse responses [10]. Mitchell et al. used comorbidities, age, and injury information to predict survival rates and ICU admission [11]. Recently, Liu and Salinas published an extensive review on how machine learning has been used in the study of trauma [12]. In general, studies have focused on the capability to predict mortality, hemorrhage, and hospital length of stay. The datasets used in these studies generally came from local trauma centers and varied greatly in training and test set size, with most studies on the order of hundreds of patients and some on the order of thousands [7, 13]. Models based on ordinary and partial differential equations have also been used to study trauma but were not used in this report [14–18].

Here, we take a machine learning approach based on a gradient boosting classifier [19] for predicting survival probabilities. Furthermore, we make use of Shapley values to garner a physiological and quantitative understanding of why patients are either at high-risk or low-risk. With a reasonably small set of 8 easily measurable and commonly known features, we demonstrate accurate prediction of patient survival probabilities and the ability to indicate patient warning signs.

## 2. Methods

### 2.1 Patient dataset

All training and testing data was obtained from the National Trauma Data Bank (NTDB), the largest aggregation of trauma data ever assembled in the United States [20]. The 2016 NTDB dataset was used for all training and testing and consisted of 968665 unique patients. Each patient was identified by a unique incident key with comorbidities, vital signs, and injury information, present in separate.csv files. The open-source library, *Pandas*, was used to import, clean, and merge each of the csv incident files and generate a matrix of features and a vector of outcomes. Input features consisted of binary categorical features (e.g., gender, alcohol use disorder, etc.) and numerical features (e.g., age, systolic blood pressure, heart rate, etc.) while the outcome vector consisted of the binary *states*, *survived or deceased*.

## 2.2 Preprocessing

Of the 968,665 unique patients in the trauma database, 351,253 patients contained missing data. The death rate of the population of patients with missing data was 1.4 times greater than that of the patients without, suggesting that patients with missing data should not be ignored. Therefore, we used an iterative imputation method (see S1 File) to impute the missing values of all patients missing 2 or fewer features. This threshold was chosen based on the distribution of quantity of missing features, which is shown in S1 Fig. This captured 181,821 additional patients, and the death rate of all included patients was now approximately equal to that of the excluded patients.

Commonly presenting categorical features (hypertension, alcoholism, etc.) that were initially present in the comorbidities.csv file were encoded into their own binary columns, indicating whether a patient had the preexisting comorbidity or not. Continuous variables such as vital signs were also included. The feature matrix, **X**, is NxM dimensional where N is the total number of patients and M is the total number of features used to construct the model. All feature values in the feature matrix were rescaled to be between 0 and 1 by the minimum and maximum of each feature, i.e.,

$$\boldsymbol{X}\_\boldsymbol{s}_{:,j} = \frac{\boldsymbol{X}_{:,j} - \min(\boldsymbol{X}_{:,j})}{\max(\boldsymbol{X}_{:,j}) - \min(\boldsymbol{X}_{:,j})}, \tag{1}$$

where $\boldsymbol{X}\_\boldsymbol{s}_{i,j}$ is the rescaled $j^{\text{th}}$ column in the new feature matrix and $\boldsymbol{X}_{:,j}$ is the unscaled $j^{\text{th}}$ column of the original feature matrix. Feature rescaling is a standard preprocessing step performed so that all features are dimensionless and of the same order of magnitude.

## 2.3 Class imbalance

One of the challenges associated with classifying whether the patient *survived* or *deceased* is that the dataset had 778303 *survived* patients and only 20930 *deceased* patients, a very large class imbalance [21]. We chose to address this problem by undersampling from the *survived* class. A total of 85% of the *deceased* patients were randomly selected to be included in the training set, and an equal number of *survived* patients were randomly selected to be included in the training set. All other patient records were included in the test set resulting in a training set of size 35580 records and a test set of size 763653 records, as shown in Fig 1B.

## 2.4 Feature selection

A critical step in determining the final accuracy and utility of the model in the trauma unit (i.e., bedside) was determining which features to include. In this study, features were chosen not only based upon which would be the most predictive of outcome but also upon which would be the easiest to measure on admission. The permutation importance (PI) method [22] was used to determine which features were most likely to be the most predictive of trauma, see Fig 1C. The method consists of training a model, obtaining an accuracy for that model on the independent test set, then randomly permuting each feature column and measuring the change in accuracy. If the accuracy of the model decreases significantly, then this implies that the permuted feature was heavily contributing to the prediction of the model and should be kept in the final model.

In this case, the PI method guided the reduction of 32 feature to only 8 final features per patient record. The final 8 features used for model training were: systolic blood pressure (SBP), heart rate (HR), respiratory rate (RR), temperature (Temp), oxygen saturation (SaO2), gender, age, and total Glasgow coma score (GCSTOT). The full list of features before and after

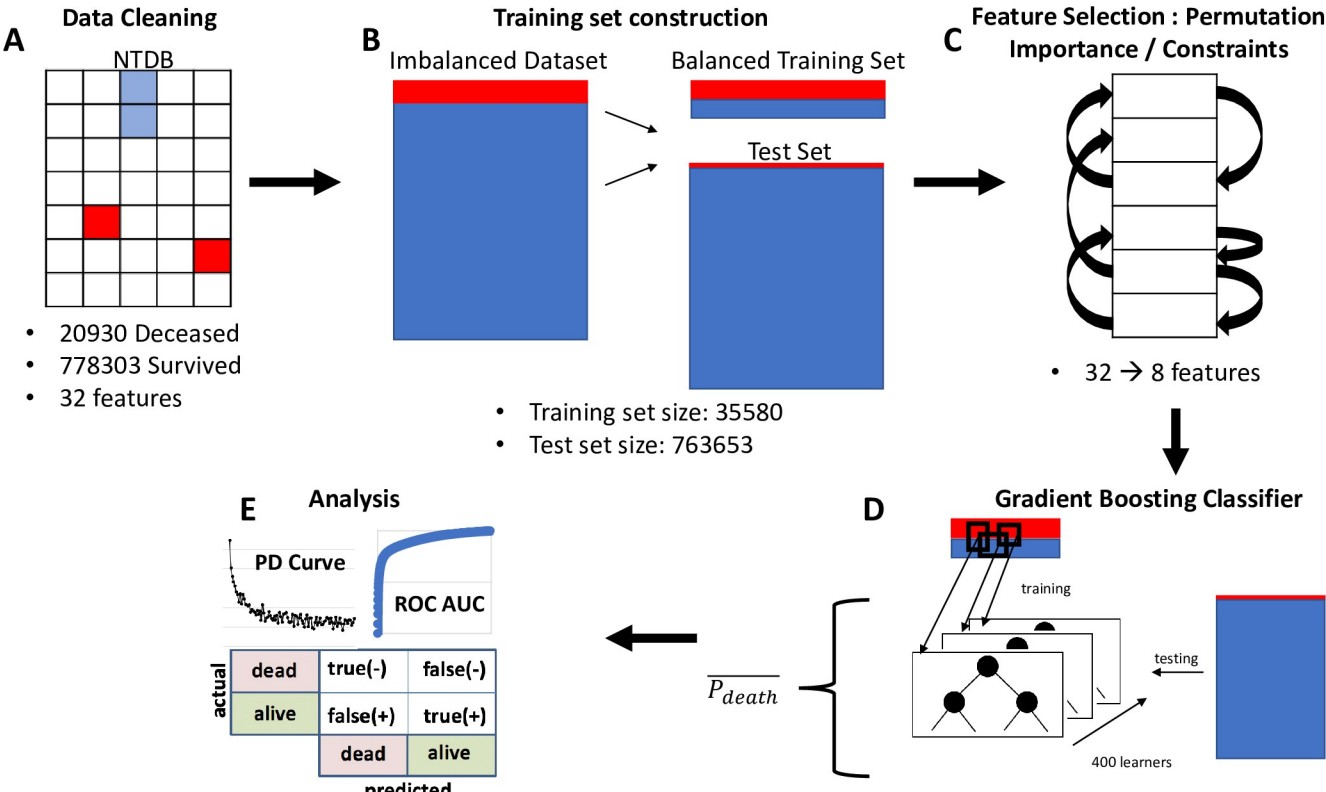

**Fig 1. Process flow diagram of the process of building a predictive trauma model.** The dataset is acquired from the National Trauma Data Bank (NTDB) and any patient with more than 2 missing data fields were removed from the dataset (cleaning). The data consisted of relational tables with each patient identified by a unique incident key. By merging using the incident key, it was possible to generate a matrix of data where each row represented a unique patient and each column represented a unique feature. Features were included in the column based on physiological information that was expected to contribute to the outcome of the model. This included age, gender, vital signs, coma and severity scores, and comorbidities. Facility and demographic information (other than age) was not included in the analysis. The dataset was then divided into a balanced training set (equal number of *survived* and *deceased* patients) and a test set, a model was trained on the training set with optimized hyperparameters (see S2 Fig), and then the results reported and analyzed.

the reduction can be found in S1 Table. While the Glasgow Coma Score can be unreliable in intubated patients, only an exceedingly small percentage (<1%) of the entire NTDB trauma patient set were in that state [23].

Our description of the gradient boosting classifier, partial dependence curves, and SHAP values can be found in S1 File [5, 24–27].

## 3. Results

The accuracy of the model is expressed as the area under the receiver operating characteristic curve coefficient (AUC). Given two patients, one *survived* and one *deceased*, the AUC represents the relative likelihood of the classifier predicting that the patient who survived had the higher probability of surviving. An AUC of 0.50 is the worst-case, as it implies that the classifier is no better than random guessing while an AUC of 1.0 is the best case, as it will always classify the two patients correctly. The AUC method is also relatively insensitive to the class imbalance between the *survived* and *deceased* patients making it a logical choice as the metric for accuracy. The gradient boosting model was able to achieve an accuracy of 0.924.

In other words, with a single 8-feature vector consisting of age, gender, five vital sign measurements, and the easily measurable Glasgow coma score, it was possible to predict the

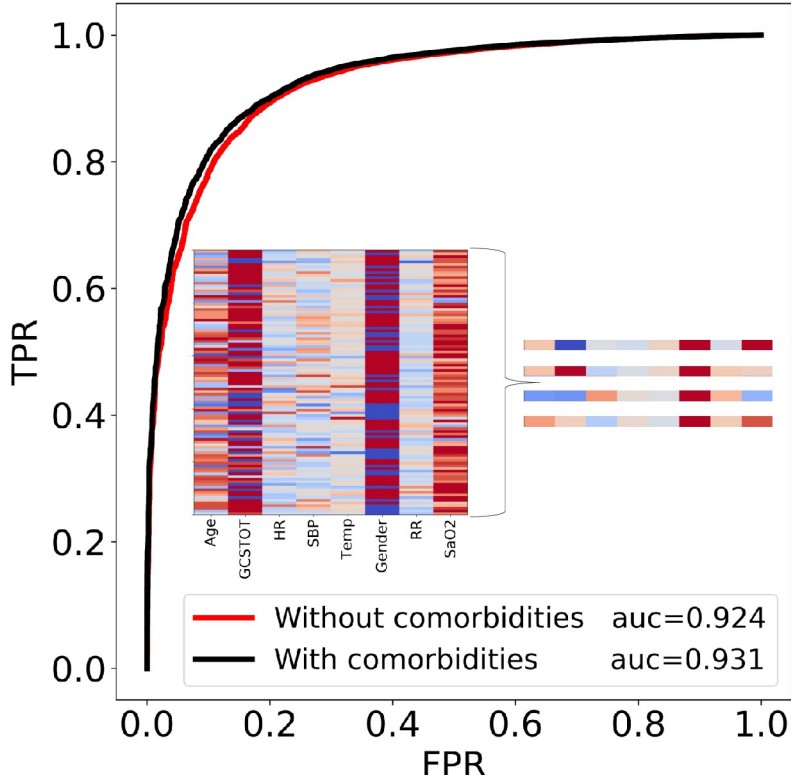

**Fig 2. The receiver operating characteristic curves (ROC) for 2 different cases.** The true positive rate (TPR) is plotted on the y-axis and the false positive rate (FPR) is plotted on the x-axis for classification thresholds between 0 and 1. In the red curve, only 8 easily measurable vital signs or scores were included in the prediction while the black curve included these and the comorbidities. A full list of features in each case can be found in S1 Table. All results are reported using the second case because the required inputs can be measured rapidly, while knowledge of the comorbidities of a patient is less likely. The heat map in the insert plots the 8 feature values of 100 randomly selected patients, illustrating the high dimensionality of the problem. While no obvious pattern can be seen by humans in the heat map, the algorithm is able to find and quantify one. 4 zoomed-in examples are provided for clarity. Note that each column is normalized by its own feature value range.

outcome of the patient up to ~92.4% accuracy, making this a useful tool for quantifying patient risk. Using 32 features per patient for model training resulted in minimal improvement of the AUC (black line, Fig 2). Importantly, the high accuracy of the model implies that a single snap-shot view (8 features) can give a quantitative prediction of the patient's mortality risk *on admission*. For further validation, we tested the model on the 2017 Trauma Quality Programs participant use file (TQP PUF). The dataset consisted of an additional 648192 complete patient records and our model was able to achieve an accuracy of 91.2%, further validating the robustness of the model and eliminating concerns of data leakage and biased evaluation. The high accuracy on a completely different cohort of patients is perhaps unsurprising given that the data points from the NTDB represent patients from numerous trauma centers. As we show below, the utility of the model is not only in its prediction of mortality risk, but also in its insight in quantifying key metrics that could be viewed as potential warning signs in a trauma setting.

The partial dependence curves are shown in Fig 3. Age, GCSTOT, and systolic blood pressure all display substantial influence on the probability of survival. The model predicts that by the age of ~60, a patient's "youth protection" has substantially dissipated on average and ages greater than this will further reduce the probability of survival. Likewise, a GCSTOT below 12

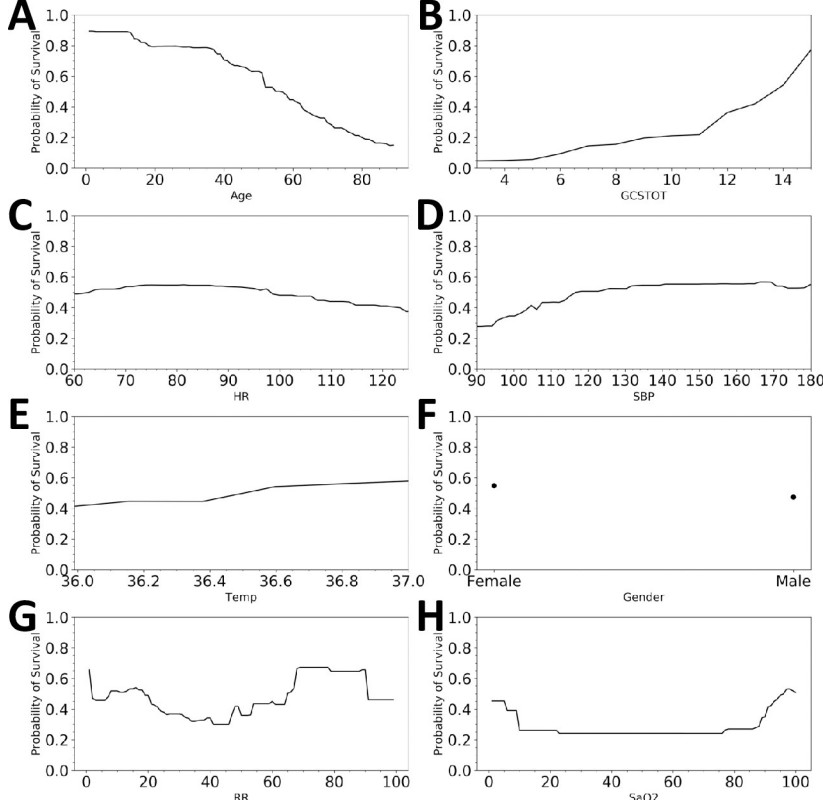

**Fig 3. Partial dependence curves showing how the prediction of the model is globally influenced by each of the features.** Pulse rate and systolic blood pressure display threshold behavior, where the probability of survival can decrease at HR > 100 beats / min and SBP < 110mmHg [30].

will increase the likelihood of death. More interestingly however, is the apparent threshold behavior in the heart rate and blood pressure profiles. The probability of survival begins to drop dramatically if the SBP < ~110 mmHg or the HR > ~100 beats per minute, consistent with the findings that hypotension is correlated with higher mortality rates [28–30]. The two variables are related to one another via the baroreflex, a negative-feedback loop system that increases heart rate in response to the loss in blood pressure, which will decline as blood volume is lost from the injury. Both vital signs should be viewed in tandem to assess patient status during resuscitation.

The probability of survival model predictions for *deceased* and *survived* patients were plotted on histograms in Fig 4 for a visual representation of the effectiveness of the model. The distributions of survival probability had means of 0.21 (*deceased*) and 0.78 (*survived*) and were highly skewed (1.51 and -1.38, respectively) suggesting that the model was very confident in the predictions that it made.

Next, SHAP values were used to examine individual patient records and quantify patient risk. As examples, 4 cases are shown in Fig 5. Note that the scales of Fig 5 are expressed as the log-odds ratio of the probability of survived to probability of deceased (i.e., $\log(\frac{p_{surv}}{1-p_{surv}})$). A log-odds ratio of 0 ($P_{surv} = 0.5$) was used to binarize the patients into *survived* and *deceased* patients. With this metric, the model correctly predicted all 4 patient outcomes in Fig 5. The force plot of the SHAP values of each feature identifies the relative contribution of each variable, both positively (blue) and negatively (red). In panel A, while the patient was conscious (GCSTOT = 15) and had relatively normal heart and respiratory rates, his low blood pressure

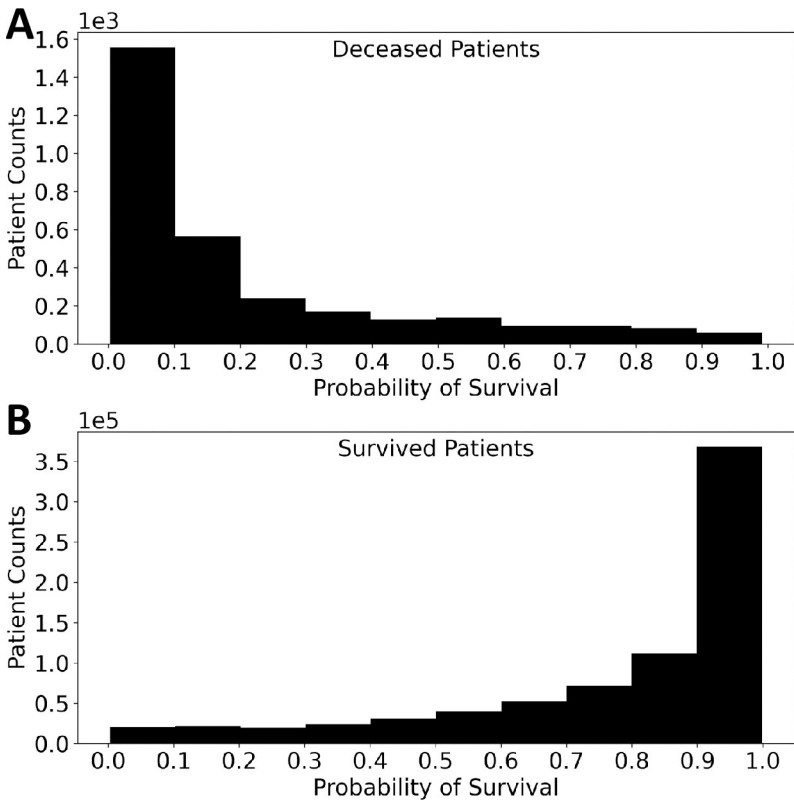

**Fig 4. Histograms of the survival probabilities for *survived* and *deceased* patients.** If probabilities of death greater than 20% are marked as high risk, then ~96% of the deceased patients would been labeled.

and age were quantifiably more significant and the reason the model predicted *deceased* (sum of SHAP values < 0). In case B, the patient's youth and consciousness were enough to overcome his abnormal vital signs. While the patient did experience mild tachypnea and tachycardia, he was ultimately not a very high-risk patient. In the third case, the patient's youth and consciousness could not compensate for the significant drop in blood pressure and elevated respiratory rate. The model identified 60 mmHg systolic blood pressure as a "red flag", which should indicate to a trauma team that this patient is a priority. In panel D, the patient's age and oxygen saturation were the key warning signs and the reason she was high-risk.

We also computed the SHAP values for 8 cases where the model made incorrect predictions, as shown in Figs 6 and 7. Notably, in all 8 cases, there was a single feature that dominated the model prediction (GCSTOT, Age, and SaO2) instead of a combination of features as in Fig 5. Machine learning models are typically best at making predictions on unseen data that are as close to an interpolation of the training data as possible, and often fail when the unseen data is significantly different from the training data. One possible solution could be to explicitly model cross terms in the data to force the model to consider all feature-feature interactions. While the model is ideally learning the feature-feature interactions during the training process, sometimes explicit inclusion of these terms can improve model accuracy (although potentially at the expense of model interpretability).

## 4. Discussion

The NTDB-trained gradient boosting model was trained with thousands of trauma patients from participating trauma centers around the country and was able to fit a robust decision

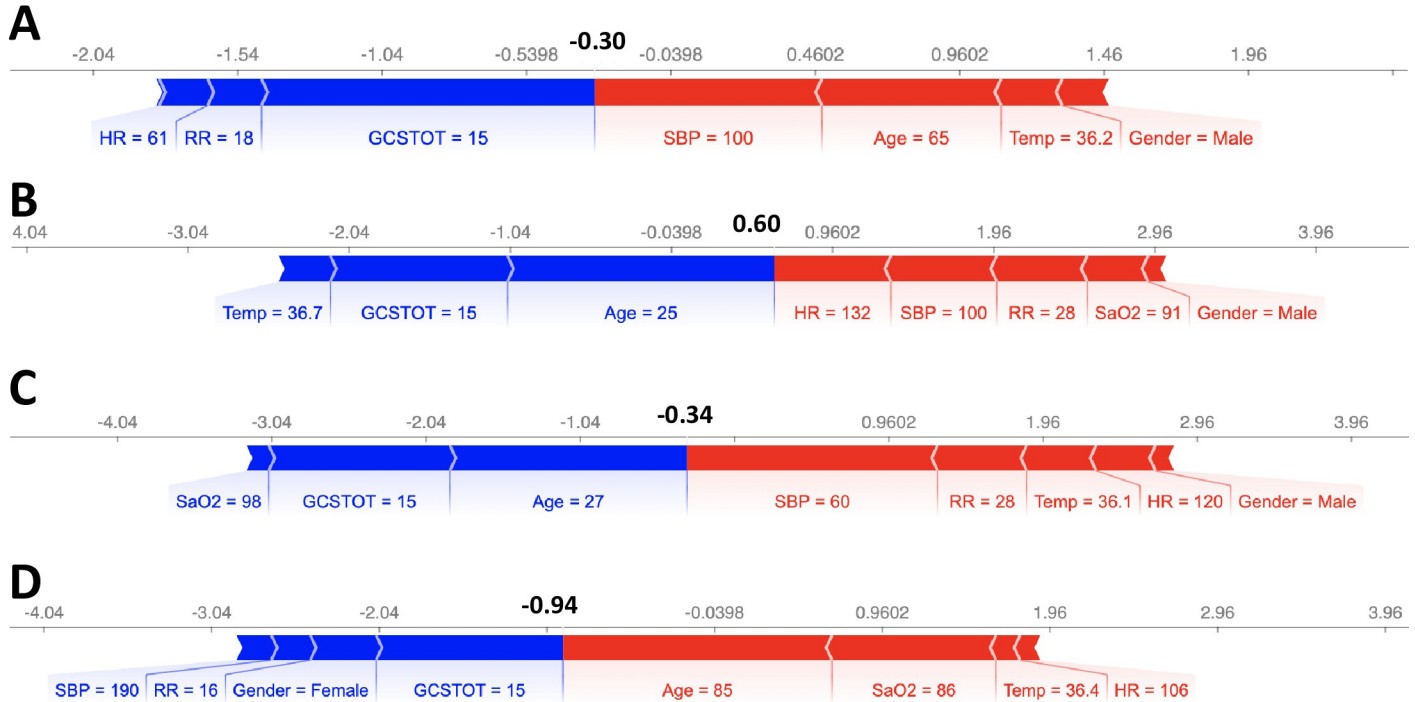

**Fig 5. SHAP feature importance metrics for 4 patients that were correctly predicted as *survived* or *deceased*.** Output values (bold), expressed as log odds ratio of probability of survival to probability of deceased (i.e. $\log(\frac{p_{surv}}{1-p_{surv}})$), that are $< 0$ represent deceased patients (Cases A, C, D). Blue bars indicate that the feature value is increasing the probability of survival while red bars indicate that the feature is decreasing it.

boundary to the dataset. With only 8 features that can be measured upon a patient's ER admission, our model was able to provide an accurate metric for patient risk of death (AUC = 0.924, Fig 2) even when death was rare. This high accuracy was further tested on a withheld dataset to further validate our claims of its high accuracy.

In a trauma setting, the usefulness of a model is not only limited by its accuracy but also by its interpretability–clinicians must understand how the model makes predictions in order to trust it. This has typically resulted in the use of linear models, but unfortunately, linear models are incapable of modeling complex decision boundaries—resulting in the loss of accuracy in exchange for interpretability. SHAP scores circumvent this problem by providing a robust method for quantitatively explaining a model's predictions. Using our model in conjunction with SHAP scores for the 8 features (Figs 5–7) provides a detailed and quantitative view of each vital sign's contribution to risk. The method may serve several distinct uses. In a prioritization setting, where both time and nurse/surgeon availability are limited, the rapid generation of a hierarchy for patient treatment has value. The model can provide an objective ranking as to which patients should receive the most available resources and guide triage. Another use is to help explain those objective rankings with specific references to patient vital signs, potentially enhancing the prioritization. Furthermore, they can be used to evaluate how actions taken to alter these variables may affect patient survival probability.

A limitation of this model is that it was trained based on the vital signs of patients upon admission. While the model was accurate, predicting the time-series evolution of the patient will requires dynamical training data. A natural extension would be to train a model that can predict patient-risk in real-time if time-series trauma patient data is available.

On the machine learning side, one limitation of our approach was the random undersampling procedure for balancing the number of *survived* patients with the number of *deceased*

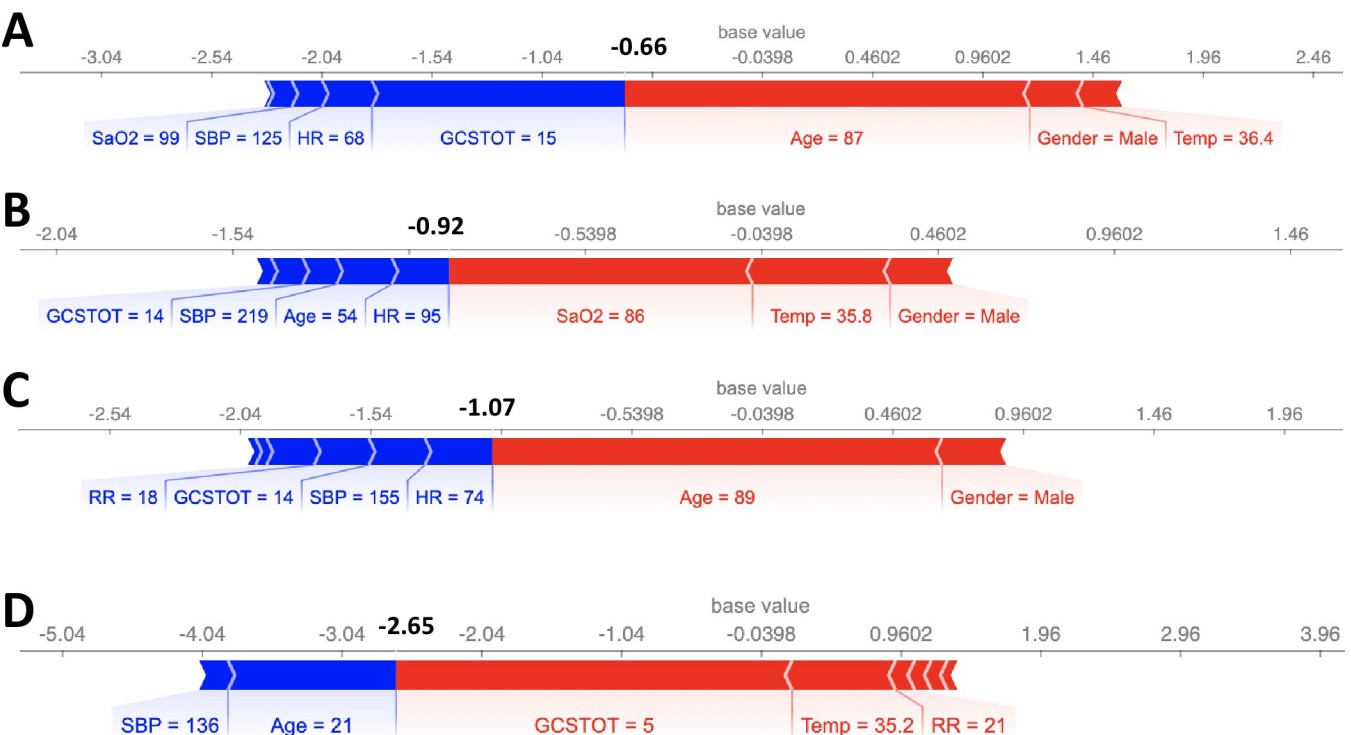

**Fig 6. SHAP feature importance metrics for 4 patients that were incorrectly predicted as deceased.** Output values (bold), expressed as log odds ratio of probability of survival to probability of deceased (i.e. $\log(\frac{p_{surv}}{1-p_{surv}})$), that are $< 0$ represent deceased patients. Blue bars indicate that the feature value is increasing the probability of survival while red bars indicate that the feature is decreasing it. In all 4 cases, there was one feature that dominated the model prediction.

patients in the training set [21]. It is possible that more informative survived patients were not included in the training set that could have led to an even more robust decision boundary [31]. Undersampling methods that include *survived* examples based upon their distances to *deceased* patients in the 8-dimensional space could improve model predictability, as they can more accurately model the decision boundary near "hard-to-classify" cases [31]. We also used the synthetic minority oversampling technique (SMOTE) to try to balance the training set, where artificial patient records from the *survived* class are generated from existing survived patient records, but it was ineffective in this instance and the accuracy of our model decreased significantly [32].

Although not the focus of this paper, we also note that the gradient boosting method exhibited a ROC-AUC that exceeded that of various neural networks and other machine learning models. Neural networks and tree-based models are two of the most commonly used classification models in the machine learning community with neural networks frequently outperforming tree-based models as well [33]. While we extensively tuned the parameters to the neural network in hopes of attaining a higher accuracy, it was unable to eclipse our gradient boosting model.

The advantages of the present approach are: (1) only 8 features are needed, (2) all 8 features are readily available on admission, (3) the calculation is exceedingly fast, portable and accurate, (4) the relative risk of each feature is determined and graphically presentable as in Figs 5 and 6, and (5) actual outcomes can be compared to the NTDB average performance on an individual basis. While features were chosen for inclusion based upon availability and importance, the accuracy of the model could be further improved by also including approximated inputs. For example, trauma surgeons generally also have an approximation for the injury severity score

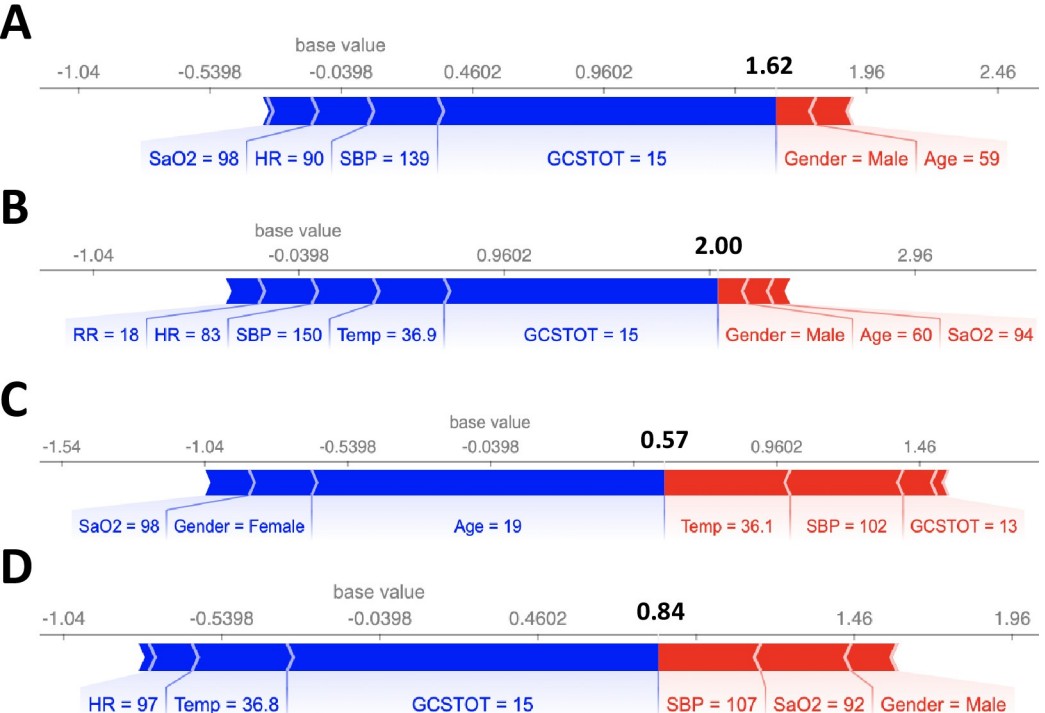

**Fig 7. SHAP feature importance metrics for 4 patients that were incorrectly predicted as survived.** Similar to incorrectly predicting deceased cases, there was one feature that dominated the model prediction.

upon admission. If estimates for the severity of each injury is included into the model, the accuracy of the model was found to increase by ~2–3%. In future work, a goal will be to determine if the model predictions can be refined as a patient's vital signs evolve in time.

## Supporting information

**S1 Fig. A distribution of the number of patients per number of missing features.** The distribution is bimodal, suggesting that including patients with a maximum of 2 missing features is the appropriate threshold for inclusion.
(TIF)

**S2 Fig. The 5-fold grid-search cross validation method for selecting the approximately optimal hyperparameters.** Importantly, the test set was withheld during the grid-search cross validation process allowing it to remain a fair metric for evaluating performance on new data.
(TIF)

**S3 Fig. A single weak learner randomly chosen from the trained gradient boosting ensemble of weak learners.** Variable thresholds, Friedman mean squared errors [19], percentage of training samples passed through each node, and log odds ratios are all present.
(TIF)

**S1 Table. Table of all features used to make predictions.**
(DOCX)

**S1 File.**
(DOCX)

## Author Contributions

**Conceptualization:** Talid Sinno, Scott L. Diamond.

**Formal analysis:** Evan J. Tsiklidis.

**Investigation:** Evan J. Tsiklidis.

**Methodology:** Evan J. Tsiklidis, Talid Sinno.

**Project administration:** Carrie Sims, Scott L. Diamond.

**Resources:** Scott L. Diamond.

**Software:** Evan J. Tsiklidis.

**Supervision:** Carrie Sims, Talid Sinno, Scott L. Diamond.

**Validation:** Evan J. Tsiklidis, Talid Sinno, Scott L. Diamond.

**Visualization:** Evan J. Tsiklidis, Talid Sinno.

**Writing – original draft:** Evan J. Tsiklidis.

**Writing – review & editing:** Evan J. Tsiklidis, Carrie Sims, Talid Sinno, Scott L. Diamond.

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
