## [Decision Letter · Decision Letter 0]

29 Jul 2020

PONE-D-20-16709

Using the National Trauma Data Bank (NTDB) and machine learning to predict trauma patient mortality at admission

PLOS ONE

Dear Dr. Diamond,

Thank you for submitting your manuscript to PLOS ONE. After careful consideration, we feel that it has merit but does not fully meet PLOS ONE’s publication criteria as it currently stands. Therefore, we invite you to submit a revised version of the manuscript that addresses the points raised during the review process.

We look forward to receiving your revised manuscript.

Kind regards,

Zsolt J. Balogh, MD, PhD, FRACS

Academic Editor

PLOS ONE

Journal Requirements:

Reviewers' comments:

Reviewer's Responses to Questions

**Comments to the Author**

1. Is the manuscript technically sound, and do the data support the conclusions?

Reviewer #1: Yes

Reviewer #2: Partly

Reviewer #3: Yes

2. Has the statistical analysis been performed appropriately and rigorously? 

Reviewer #1: I Don't Know

Reviewer #2: I Don't Know

Reviewer #3: Yes

3. Have the authors made all data underlying the findings in their manuscript fully available?

Reviewer #1: Yes

Reviewer #2: No

Reviewer #3: Yes

4. Is the manuscript presented in an intelligible fashion and written in standard English?

Reviewer #1: Yes

Reviewer #2: Yes

Reviewer #3: Yes

5. Review Comments to the Author

Reviewer #1: I congratulate the authors for using the NTDB and TQIP in their research efforts. I cannot comment on the methodology. I would ask the authors to expand their discussion on when the model predicted death, but the patients survived. I will ask the authors on how they think clinicians can use this methodology. Is in selecting the right research patients? can this method be used prehospital to guide triage to the appropriate location? will it be used to limit care? these issues are important, and as the authors have launched down this pathway of clinical decision support (which I personally support) they must marry the technology with how we are supposed to ethically utilize.

GCS is notoriously unreliable when patients are intubated. how did the authors handle this? Lastly, I agree that the traditional cut points for HR and SBP are wrong, as shown by Eastridge el al. over a decade ago.

Eastridge BJ, Salinas J, McManus JG, et al. Hypotension begins at 110 mm Hg: redefining "hypotension" with data. 2008 Aug;65(2):501. Concertino, Victor A. Trauma. 2007;63(2):291-299.

Reviewer #2: The authors have used an immense dataset (nearly 1 million patients) from the USA National Trauma Data Bank to generate a prediction model for risk of death from major trauma.

Broadly, the article provides quantitative support for the ability of an experienced observer to pick the ‘gestalt’ of a patient. Those experienced surgeons who can pick "big sick" just by seeing the patient in the trauma bay with the most basic of clinical observations.

Specific comments

The authors excluded incomplete data due to their concern regarding introduction of bias. 36% of the initial dataset was excluded due to incomplete data. The authors tantalizingly mention that the excluded data had over double the risk of death compared to the included data. As is often the case, these patients may be the most interesting, and is the real weakness with large database studies like this. There is no analysis beyond the relative mortality rate of the included vs excluded groups. The authors should explore the characteristics of the missing data more. It is very surprising that the authors have access to mortality data from the incomplete records, but not to admission HR/RR/GCSTOT/SBP/Age/Temp/Gender which could greatly expand the test dataset and reassure the reader that the model doesn’t lack external validity.

The authors allay concerns about this exclusion weakening the external validity of the model by further testing it against the TQP PUF dataset from 2017 which is a separate population. The model maintains predictive power of over 90% in this dataset which is reassuring.

The authors explore the characteristics of patients the model incorrectly overclassified as likely to die, but did not explore those under-classified. From a trauma systems point of view (where trauma activation criteria is deliberately over-inclusive, to avoid missing at-risk patients), it would be very useful to understand the characteristics of patients whom the model failed to identify as likely to die. This is where the real risk lies in implementing a model like this as a clinical decision aid. (Ambulance dispatch, remote and regional hospital transfer decisions etc)

The article’s discussion section could be expanded to discuss the usefulness of the model’s predictive ability. There should be a structured discussion regarding limitations of the study.

Line 308 is unhelpful and should be revised.

“Clinicians continually make decisions based on complex information, however they do not construct highly predictive, non-linear functions, in an 8-dimensional space based upon statistical learning over 10^4 to 10^5 cases.”

It can be argued that clinicians actually do exactly this (albeit over 10^3 – 10^4 patients instead) and that that’s what clinical experience and training is. One potential revision is to point out that models like this allow ‘experience’ to be generated programmatically, allowing it to be applied in personnel-limited settings and over many more patients than any one clinician can establish in their experience.

Reviewer #3: In this article the authors try to estimate mortality by utilizing a machine learning modality, using the National Trauma Data Bank. In a final setup 8 variables were identified that could further detail risk patterns for mortality.

Research question and embedding in current literature: the rationale and timeliness are adequate and very well delineated in the introduction.

Methods used: adequate methods were used, with separate modeling, training and evaluation sets. The sample set with almost 1 million unique datasets is a very good basis for this kind of studies. One can considerably argue about the removal of incomplete datasets. Even more if the death rate in this set is 2.5 times higher than that for complete sets. Not being a advocate for imputation, however leaving out a that kind of skewed population comes with hazards for the outcome, which should be better substantiated. Did the authors consider a test run with and with out imputation to at least have a faint idea what the consequences are of this way of treating the data?

Moreover the authors decided to go for a random sample for the training database and because of the skewness toward non-deceased under sampled the non-survivors. Is there any idea on what the consequences were? Any time differences? Adequacy of the randomization proces? How did this under sampling take technically place?

In the set the GCS is taken as an important factor. One of the major problems with the GCS is that it can be influenced by the pre-hospital treatment and is especially problematic in the severely injured patients that come in already intubated. How did the authors correct for this?

Why was chosen for the supervised learning method as this already points mostly into a certain direction of the results. Any comment on this?

The explanation of the machine learning proces (from line 151 on) is very detailed, however adequate, but it possibly would be better to get this part into the supplemental material as most of the PLOS readership will not have any connection with it. From paragraph 2.6 could be left in.

Results: the results are impressive with an accuracy of over 91 %.The authors advise that the vital signs (BP and HR) should be viewed as tandem, mainly because these are interrelated. Could they explain how they would see (advice) this in practice. This is in line with the very old Algöwer principal of the shock index.

In line 265 and further the authors should explain a little bit more what they mean with their statement, taken the average readership in mind.

Discussion: The authors constantly have the average clinician in mind and should be complimented with that, however no real advise how to use the very nice findings in clinical practice can be found in the discussion. How could the readership get advantage from the findings that are obviously better than most predictions at hand, but how should they going to use it in clinical practice what are the next steps to bring this to the doctor and the patient, who should benefit from this.

Moreover in the discussion is very limited and no real comparison with the literature has been made, which should be extended in my mind.

6. PLOS authors have the option to publish the peer review history of their article (what does this mean?). If published, this will include your full peer review and any attached files.

Reviewer #1: No

Reviewer #2: No

Reviewer #3: No

---

## [Author Response · Author response to Decision Letter 0]

15 Aug 2020

Response to reviewers document included in the files. However, a copy and pasted version is available below. 

Response to Reviewer #1 (Our comments in bold and italics)

Reviewer #1: I congratulate the authors for using the NTDB and TQIP in their research efforts. I cannot comment on the methodology. I would ask the authors to expand their discussion on when the model predicted death, but the patients survived. 

We thank the reviewer for their supportive comments. 

1. I will ask the authors on how they think clinicians can use this methodology. Is in selecting the right research patients? Can this method be used prehospital to guide triage to the appropriate location? will it be used to limit care? these issues are important, and as the authors have launched down this pathway of clinical decision support (which I personally support) they must marry the technology with how we are supposed to ethically utilize.

We have added:

The method may serve several distinct uses. In a prioritization setting, where both time and nurse/surgeon availability are limited, the rapid generation of a hierarchy for patient treatment has value. The model can provide an objective ranking as to which patients should receive the most available resources and guide triage. Another use is to help explain those objective rankings with specific references to patient vital signs, potentially enhancing the prioritization. 

2. GCS is notoriously unreliable when patients are intubated. How did the authors handle this? Lastly, I agree that the traditional cut points for HR and SBP are wrong, as shown by Eastridge el al. over a decade ago.

Eastridge BJ, Salinas J, McManus JG, et al. Hypotension begins at 110 mm Hg: redefining "hypotension" with data. 2008 Aug;65(2):501. Concertino, Victor A. Trauma. 2007;63(2):291-299.

We agree that GCS can be unreliable when patients are intubated and have included some context regarding intubation. Of the 799,233 patient population, only 903 patients were labeled as “intubated or chemically sedated”, representing only ~0.1% of the population, a likely very small contribution. We added the statement: 

“While the Glasgow Coma Score can be unreliable in intubated patients [Meredith], only an exceedingly small percentage (<1%) of the entire NTDB trauma patient set were in that state.”

Meredith, W., Rutledge, R., Fakhry, S. M., Emery, S., & Kromhout-Schiro, S. (1998). The conundrum of the Glasgow Coma Scale in intubated patients: a linear regression prediction of the Glasgow verbal score from the Glasgow eye and motor scores. The Journal of trauma, 44(5), 839–845. https://doi.org/10.1097/00005373-199805000-00016

We also included the earlier references identified by the reviewer in our discussion on the threshold behavior of the systolic blood pressure of trauma patients. 

 

Response to Reviewer #2 (Our comments in bold and italics)

Reviewer #2: The authors have used an immense dataset (nearly 1 million patients) from the USA National Trauma Data Bank to generate a prediction model for risk of death from major trauma. Broadly, the article provides quantitative support for the ability of an experienced observer to pick the ‘gestalt’ of a patient. Those experienced surgeons who can pick "big sick" just by seeing the patient in the trauma bay with the most basic of clinical observations.

We thank the reviewer for their thoughts. 

Specific comments

1. The authors excluded incomplete data due to their concern regarding introduction of bias. 36% of the initial dataset was excluded due to incomplete data. The authors tantalizingly mention that the excluded data had over double the risk of death compared to the included data. As is often the case, these patients may be the most interesting, and is the real weakness with large database studies like this. There is no analysis beyond the relative mortality rate of the included vs excluded groups. The authors should explore the characteristics of the missing data more. It is very surprising that the authors have access to mortality data from the incomplete records, but not to admission HR/RR/GCSTOT/SBP/Age/Temp/Gender which could greatly expand the test dataset and reassure the reader that the model doesn’t lack external validity. The authors allay concerns about this exclusion weakening the external validity of the model by further testing it against the TQP PUF dataset from 2017 which is a separate population. The model maintains predictive power of over 90% in this dataset which is reassuring.

The decision to remove the missing data records was difficult but necessary. Motivated by the reviewer comment, we have gone back and taken a closer look at the data that was removed. We observed that the death rate was 1.4x higher in the missing data group than in the complete data group, which was too significant to ignore. Therefore, imputation was performed on missing entries for patients who were missing a maximum of 2 features. This increased the size of the usable patient database to 799,233 patients and the accuracy increased to ~92.4%. We thank the reviewer for this suggestion. Most notably, the death rate of the remaining excluded data (with more than 2 missing features) was now found to be approximately equal to that of the included data – which should further allay concerns over excluded data. This is now discussed further in the manuscript.

In the methods section, we added: 

Of the 968,665 unique patients in the trauma database, 351,253 patients contained missing data. The death rate of the population of patients with missing data was 1.4x greater than that of the patients without, suggesting that missing data could not be ignored. Therefore, we used an iterative imputation method (see supplemental section for details) to impute the missing values of all patients missing 2 or fewer features. This captured 181,821 additional patients, and the death rate of all included patients was now approximately equal to that of the excluded patients. 

In the supplemental section, we added: 

We used the IterativeImputer class from the scikit-learn library to impute missing data in patients missing 2 or fewer features [Pedregosa]. Missing features were modeled as functions of present features and a ridge linear regression model was trained to predict the missing value. At a single step in the iteration, a single feature was treated as the missing output while the remaining features were assigned as inputs. This was repeated for each feature in a single round, and then iterated for 50 rounds. In the rare instances that this imputation led to unphysical values (e.g., age < 0), we simply imputed the value with a nearby physical value. This is superior to simply imputing the missing value with the mean of the present features as it imputes the average over many approximations of possible values. Furthermore, simple imputation significantly model variance leading to lower generalizability. 

2. The authors explore the characteristics of patients the model incorrectly overclassified as likely to die, but did not explore those under-classified. From a trauma systems point of view (where trauma activation criteria is deliberately over-inclusive, to avoid missing at-risk patients), it would be very useful to understand the characteristics of patients whom the model failed to identify as likely to die. This is where the real risk lies in implementing a model like this as a clinical decision aid. (Ambulance dispatch, remote and regional hospital transfer decisions etc)

An additional figure (Fig. 7) and discussion for this under-classified case has been added to the paper. We have added/edited the following from the discussion section: 

We also computed the SHAP values for 8 cases where the model made incorrect predictions, as shown in Fig. 6 and Fig. 7. Notably, in all 8 cases, there was a single feature that dominated the model prediction (GCSTOT, Age, and SaO2) instead of a combination of features as in Fig. 5. Machine learning models are typically best at making predictions on unseen data that are as close to an interpolation of the training data as possible, and often fail when the unseen data is significantly different from the training data. One possible solution could be to explicitly model cross terms in the data to force the model to consider all feature-feature interactions. While the model is ideally learning the feature-feature interactions during the training process, sometimes explicit inclusion of these terms can improve model accuracy (although potentially at the expense of model interpretability).

3. The article’s discussion section could be expanded to discuss the usefulness of the model’s predictive ability. There should be a structured discussion regarding limitations of the study.

Line 308 is unhelpful and should be revised.

“Clinicians continually make decisions based on complex information, however they do not construct highly predictive, non-linear functions, in an 8-dimensional space based upon statistical learning over 10^4 to 10^5 cases.” It can be argued that clinicians actually do exactly this (albeit over 10^3 – 10^4 patients instead) and that that’s what clinical experience and training is. One potential revision is to point out that models like this allow ‘experience’ to be generated programmatically, allowing it to be applied in personnel-limited settings and over many more patients than any one clinician can establish in their experience.

We agree. The discussion section has been expanded/revised to the following: 

The NTDB-trained gradient boosting model was trained with thousands of trauma patients from participating trauma centers around the country and was able to fit a robust decision boundary to the dataset. With only 8 features that can be measured upon a patient’s ER admission, our model was able to provide an accurate metric for patient risk of death (AUC = 0.924, Fig. 2) even when death was rare. This high accuracy was further tested on a withheld dataset to further validate our claims of its high accuracy. 

In a trauma setting, the usefulness of a model is not only limited by its accuracy but also by its interpretability – clinicians must understand how the model makes predictions in order to trust it. This has typically resulted in the use of linear models, but unfortunately, linear models are incapable of modeling complex decision boundaries - resulting in the loss of accuracy in exchange for interpretability. SHAP scores circumvent this problem by providing a robust method for quantitatively explaining a model’s predictions. Using our model in conjunction with SHAP scores for the 8 features (Fig. 5-7) provides a detailed and quantitative view of each vital sign’s contribution to risk. The method may serve several distinct uses. In a prioritization setting, where both time and nurse/surgeon availability are limited, the rapid generation of a hierarchy for patient treatment has value. The model can provide an objective ranking as to which patients should receive the most available resources and guide triage. Another use is to help explain those objective rankings with specific references to patient vital signs, potentially enhancing the prioritization. Furthermore, they can be used to evaluate how actions taken to alter these variables may affect patient survival probability.

A limitation of this model is that it was trained based on the vital signs of patients upon admission. While the model was accurate, predicting the time-series evolution of the patient will requires dynamical training data. A natural extension would be to train a model that can predict patient-risk in real-time if time-series trauma patient data is available. 

On the machine learning side, one limitation of our approach was the random undersampling procedure for balancing the number of survived patients with the number of deceased patients in the training set [Liu]. It is possible that more informative survived patients were not included in the training set that could have led to an even more robust decision boundary [Zhang]. Undersampling methods that include survived examples based upon their distances to deceased patients in the 8-dimensional space could improve model predictability [Zhang], as they can more accurately model the decision boundary near “hard-to-classify” cases. We also used the synthetic minority oversampling technique (SMOTE) to try to balance the training set, where artificial patient records from the survived class are generated from existing survived patient records, but it was ineffective in this instance and the accuracy of our model decreased significantly [Fernandez]. 

Although not the focus of this paper, we also note that the gradient boosting method exhibited a ROC-AUC that exceeded that of various neural networks and other machine learning models. Neural networks and tree-based models are two of the most commonly used classification models in the machine learning community with neural networks frequently outperforming tree-based models as well [Haldar]. While we extensively tuned the parameters to the neural network in hopes of attaining a higher accuracy, it was unable to eclipse our gradient boosting model. 

The advantages of the present approach are: (1) only 8 features are needed, (2) all 8 features are readily available on admission, (3) the calculation is exceedingly fast, portable and accurate, (4) the relative risk of each feature is determined and graphically presentable as in Fig. 5 and 6, and (5) actual outcomes can be compared to the NTDB average performance on an individual basis. While features were chosen for inclusion based upon availability and importance, the accuracy of the model could be further improved by also including approximated inputs. For example, trauma surgeons generally also have an approximation for the injury severity score upon admission. If estimates for the severity of each injury is included into the model, the accuracy of the model was found to increase by ~2-3%. In future work, a goal will be to determine if the model predictions can be refined as a patient’s vital signs evolve in time. 

Fernandez A, Garcia S, Herrera F, Chawla, V. N. SMOTE for Learning from Imbalanced Data: Progress and Challenges, Marking the 15-year Anniversary. J Artif Intell Res [Internet]. 2018;61:863–905. 

Zhang J, Mani I. kNN Approach to Unbalanced Data Distributions: A Case Study involving information Extraction. In: Proceeding of International Conference on Machine Learning (ICML 2003), Workshop on Learning from Imbalanced Data Sets. 2003.

Liu X, Wu J, Zhou Z. Exploratory Undersampling for Class-Imbalance Learning.

IEEE Trans Syst Man, Cybern Part B. 2009;39(2):539–50.

Haldar M, Abdool M, Ramanathan P, Xu T, Yang S, Duan H, et al. Applying deep learning to Airbnb search. arXiv e-prints. 2018;1927-35. 

 

Response to Reviewer #3 (Our comments in bold and italics)

Reviewer #3: In this article the authors try to estimate mortality by utilizing a machine learning modality, using the National Trauma Data Bank. In a final setup 8 variables were identified that could further detail risk patterns for mortality.

Research question and embedding in current literature: the rationale and timeliness are adequate and very well delineated in the introduction.

We thank the reviewer for their thoughts.

1. Methods used: adequate methods were used, with separate modeling, training and evaluation sets. The sample set with almost 1 million unique datasets is a very good basis for this kind of studies. One can considerably argue about the removal of incomplete datasets. Even more if the death rate in this set is 2.5 times higher than that for complete sets. Not being a advocate for imputation, however leaving out a that kind of skewed population comes with hazards for the outcome, which should be better substantiated. Did the authors consider a test run with and with out imputation to at least have a faint idea what the consequences are of this way of treating the data?

We have taken a closer look at the missing data and noticed some near duplicate entries where the patient vital signs were sometimes counted twice, so when we removed them from the database – the true death rate of the excluded data population was found to actually be ~1.4 fold higher (not 2.5 fold higher). This has been corrected. However, the referee’s point is still valid and must be addressed carefully.

Handling missing data requires imputation of the missing values but it is impossible to determine the extent to which this is helping or hurting the model’s prediction (Jakobsen, Sterne). Poor imputation can add noise to the model and decrease model accuracy, while excluding missing data from the analysis can introduce bias. Ultimately, the onus is on the data analyst to consider the situation and access the missingness and propose a solution for it. 

Based on the referee’s comments, we spent considerable time on this question and came up with a solution. Of the subset of missing data, ~52% of them were missing fewer than 3 features. We chose to add this subset to the dataset by imputing the missing values with the mean value. This captured 181,821 additional patients, 7,011 of whom were deceased. We believe this is the appropriate compromise between including additional data and adding noise into the model. Furthermore, the distribution of quantity of missing features was bimodal, making a maximum of 2 missing features a natural threshold. Conveniently, this raises the death rate in the usable patient dataset to 2.62% and lowers the death rate in the excluded patient database to 2.59% - which should allay concerns of biasing. Moreover, the accuracy of the model was found to increase to 92.4% with the additional data. We thank the reviewer for this suggestion. 

The text has been updated in all relevant places to account for this change. 

In the methods section, we added: 

Of the 968,665 unique patients in the trauma database, 351,253 patients contained missing data. The death rate of the population of patients with missing data was 1.4 times greater than that of the patients without, suggesting that missing data could not be ignored. Therefore, we used an iterative imputation method (see supplemental section for details) to impute the missing values of all patients missing 2 or fewer features. This captured 181,821 additional patients, and the death of the included patients was approximately equal to that of the excluded patients. 

In the supplemental section, we added: 

We used the IterativeImputer class from the scikit-learn library to impute missing data in patients missing 2 or fewer features [Pedregosa]. Missing features are modeled as functions of present features and a ridge linear regression model is trained to predict the missing value. At a single step in the iteration, a single feature is treated as the missing output while the remaining features are the input. This is repeated for each feature in a single round, and then iterated for 50 rounds. In the rare instances that this imputation led to unphysical values (e.g., age < 0), we simply imputed the value with a nearby physical value. This is superior to simply imputing the missing value with the mean of the present features as it imputes the average over many approximations of possible values. Furthermore, simple imputation significantly model variance leading to lower generalizability. 

Jakobsen JC, Gluud C, Wetterslev J, Winkel P. When and how should multiple imputation be used for handling missing data in randomised clinical trials - A practical guide flowcharts. BMC Med Res Methodology. 2017;17(1):1–10. 2009;339(7713):157–60.

Sterne JAC, White IR, Carlin JB, Spratt M, Royston P, Kenward MG, et al. Multiple imputation for missing data in epidemiological and clinical research: Potential and pitfalls. BMJ. 2009;339(7713):157-60.

Pedregosa F, Varoquaux G, Gramfort A, Michel V, Thirion B, Grisel O, et al. Scikit-learn: Machine Learningi in Python. J Mach Learn Res. 2011;12: 2825-30

2. Moreover the authors decided to go for a random sample for the training database and because of the skewness toward non-deceased under sampled the non-survivors. Is there any idea on what the consequences were? Any time differences? Adequacy of the randomization proces? How did this under sampling take technically place?

Under sampling from the majority class to create a balanced training set is a well-established, common solution to the class-imbalance problem (Liu). However, it does come with some potential consequences that should be acknowledged. The first is that we are limiting instances of the survived population from the training set, which could be important to fitting a robust decision boundary. We may not be preserving information-rich survived patients in the training set – which could be determinantal to model fit and reduce accuracy. An alternative approach is oversampling, whereby instances from the minority class are duplicated to balance the relative number of survived – deceased patients, but this actually led to a significant reduction in accuracy. 

The randomization process was carried out by randomly selecting 85% of the deceased patients from the database to be included in the training set, and then selecting an equivalent number of randomly selected survived patients to balance the training set. The randomization process was performed with the python library, numpy, with a specified seed for reproducibility from run-to-run. 

We added: 

On the machine learning side, one limitation of our approach was the random undersampling procedure for balancing the number of survived patients with the number of deceased patients in the training set [Liu]. It is possible that more informative survived patients were not included in the training set that could have led to an even more robust decision boundary [Zhang]. Undersampling methods that include survived examples based upon their distances to deceased patients in the 8-dimensional space could improve model predictability [Zhang], as they can more accurately model the decision boundary near “hard-to-classify” cases. We also used the synesthetic minority oversampling technique (SMOTE) to try to balance the training set, where artificial patient records from the survived class are generated from existing survived patient records, but it was ineffective in this instance and the accuracy of our model decreased significantly [Fernandez]. 

Fernandez A, Garcia S, Herrera F, Chawla, V. N. SMOTE for Learning from Imbalanced Data: Progress and Challenges, Marking the 15-year Anniversary. J Artif Intell Res [Internet]. 2018;61:863–905. 

Zhang J, Mani I. kNN Approach to Unbalanced Data Distributions: A Case Study involving information Extraction. In: Proceeding of International Conference on Machine Learning (ICML 2003), Workshop on Learning from Imbalanced Data Sets. 2003.

Liu X, Wu J, Zhou Z. Exploratory Undersampling for Class-Imbalance Learning.

IEEE Trans Syst Man, Cybern Part B. 2009;39(2):539–50.

3. In the set the GCS is taken as an important factor. One of the major problems with the GCS is that it can be influenced by the pre-hospital treatment and is especially problematic in the severely injured patients that come in already intubated. How did the authors correct for this?

This comment was also made in point 2 by reviewer #1, please see our specific response there. 

4. Why was chosen for the supervised learning method as this already points mostly into a certain direction of the results. Any comment on this?

Gradient Boosting was chosen as the method as it was the most accurate at classifying the results. Unsupervised learning was performed upon the dataset, but no meaningful inferences could be made from that analysis and it was not included in the paper. 

5. The explanation of the machine learning proces (from line 151 on) is very detailed, however adequate, but it possibly would be better to get this part into the supplemental material as most of the PLOS readership will not have any connection with it. From paragraph 2.6 could be left in.

We have moved these sections to the supplemental section. 

6. Results: the results are impressive with an accuracy of over 91 %.The authors advise that the vital signs (BP and HR) should be viewed as tandem, mainly because these are interrelated. Could they explain how they would see (advice) this in practice. This is in line with the very old Algöwer principal of the shock index.

In line 265 and further the authors should explain a little bit more what they mean with their statement, taken the average readership in mind. Discussion: The authors constantly have the average clinician in mind and should be complimented with that, however no real advise how to use the very nice findings in clinical practice can be found in the discussion. How could the readership get advantage from the findings that are obviously better than most predictions at hand, but how should they going to use it in clinical practice what are the next steps to bring this to the doctor and the patient, who should benefit from this.

Lines 265 and beyond primarily discuss our insights from the SHAP value analysis. We added: 

Using our model in conjunction with SHAP scores for the 8 features (Fig. 5 and 6) provides a detailed and quantitative view of each vital sign’s contribution to risk. The method may serve several distinct uses. In a prioritization setting, where both time and nurse/surgeon availability are limited, the rapid generation of a hierarchy for patient treatment has value. The model can provide an objective ranking as to which patients should receive the most available resources and guide triage. Another use is to help explain those objective rankings with specific references to patient vital signs, potentially enhancing the prioritization. Furthermore, they can be used to evaluate how actions taken to alter these variables may affect patient survival probability.

8. Moreover in the discussion is very limited and no real comparison with the literature has been made, which should be extended in my mind.

We agree and addressed it in our response to the 3rd point made by reviewer #2. Please see our response there.

---

## [Decision Letter · Decision Letter 1]

28 Oct 2020

Using the National Trauma Data Bank (NTDB) and machine learning to predict trauma patient mortality at admission

PONE-D-20-16709R1

Dear Dr. Diamond,

We’re pleased to inform you that your manuscript has been judged scientifically suitable for publication and will be formally accepted for publication once it meets all outstanding technical requirements.

Kind regards,

Zsolt J. Balogh, MD, PhD, FRACS

Academic Editor

PLOS ONE

Additional Editor Comments (optional):

Thank you.

Reviewers' comments:

Reviewer's Responses to Questions

**Comments to the Author**

1. If the authors have adequately addressed your comments raised in a previous round of review and you feel that this manuscript is now acceptable for publication, you may indicate that here to bypass the “Comments to the Author” section, enter your conflict of interest statement in the “Confidential to Editor” section, and submit your "Accept" recommendation.

Reviewer #1: All comments have been addressed

Reviewer #2: All comments have been addressed

Reviewer #3: All comments have been addressed

2. Is the manuscript technically sound, and do the data support the conclusions?

Reviewer #1: Yes

Reviewer #2: Yes

Reviewer #3: Yes

3. Has the statistical analysis been performed appropriately and rigorously? 

Reviewer #1: Yes

Reviewer #2: I Don't Know

Reviewer #3: Yes

4. Have the authors made all data underlying the findings in their manuscript fully available?

Reviewer #1: Yes

Reviewer #2: No

Reviewer #3: Yes

5. Is the manuscript presented in an intelligible fashion and written in standard English?

Reviewer #1: Yes

Reviewer #2: Yes

Reviewer #3: Yes

6. Review Comments to the Author

Reviewer #1: the authors have adequately addressed all concerns. no additional comments for the authors. no additional comments for the authors. the authors have adequately addressed all concerns.

Reviewer #2: (No Response)

Reviewer #3: the authors have followed largely the comments mad and have changed and added considerable parts of manuscript accordingly

7. PLOS authors have the option to publish the peer review history of their article (what does this mean?). If published, this will include your full peer review and any attached files.

Reviewer #1: No

Reviewer #2: No

Reviewer #3: **Yes: **Luke PH Leenen

---

## [Editor Report · Acceptance letter]

30 Oct 2020

PONE-D-20-16709R1 

Using the National Trauma Data Bank (NTDB) and machine learning to predict trauma patient mortality at admission 

Dear Dr. Diamond:

I'm pleased to inform you that your manuscript has been deemed suitable for publication in PLOS ONE. Congratulations! Your manuscript is now with our production department. 

Kind regards, 

on behalf of

Dr. Zsolt J. Balogh 

Academic Editor

PLOS ONE